# Identification of the Visually Prominent Gait Parameters for Forensic Gait Analysis

**DOI:** 10.3390/ijerph19042467

**Published:** 2022-02-21

**Authors:** Daewook Lee, Jiman Soon, Gyuri Choi, Kijoon Kim, Sangwoo Bahn

**Affiliations:** Industrial and Management Systems Engineering, College of Engineering, Kyung Hee University, Yongin 17104, Korea; eodnr9486@khu.ac.kr (D.L.); james8134@khu.ac.kr (J.S.); cjl0205@khu.ac.kr (G.C.); karllen@khu.ac.kr (K.K.)

**Keywords:** gait, gait parameters, forensic gait analysis, walking pattern

## Abstract

Walking patterns can be used as a key parameter in identifying individuals, as it varies visually depending on one’s body size as well as their habits, gender, and age group. In this study, we measure the gait characteristics of a large number of subjects using 34 visual parameters to identify significant parameters that can be used to distinguish individual walking features. We recorded 291 subjects’ walking on a constructed footpath using four video cameras, and data on parameters was calculated at the points of double support, toe-off, and heel-strike. K-means Clustering Analysis and ANOVA were conducted to determine the difference between age, gender, and BMI. As a result, we confirm that parameters related to the spine, neck, and feet are useful for identifying individuals. In the comparative analysis between age groups, the older the age, the more significant variables appeared in the upper body. The difference between genders showed significant parameters in both the upper and lower bodies of males. Similarly, among the large BMI groups, we also derived significant results in the upper and lower bodies. The key parameters derived from this study can be used more effectively in the real-world visual analysis of gait, as the walking characteristics of a large number of subjects have been measured with a similar view as real-world CCTV. This study will be effectively utilized as a foundation for future research attempting to identify people through their gait by distinguishing major gait characteristic differences.

## 1. Introduction

The combination of different parameters in gait varies widely from individual to individual [1], and it also varies with body size, habits, gender, and age [2]. Thus, gait can be a useful factor for individual identification. Furthermore, it is difficult for an individual to hide or change their walking pattern, because any intentional changes to gait pattern are temporary [3]. Hence, gait pattern is likely to become useful in forensic identification. Accordingly, in recent years, various research methods have attempted to use gait analysis for personal identification.

Existing gait analysis and research can be divided into two main categories: gait analysis using sensors and measuring equipment, and gait characteristic extraction using video footages. For gait studies using sensors and measuring equipment, Yu et al. proposed a method to classify people’s gender using a support vector machine (SVM); they tested this method with 31 subjects and recorded a Correct Classification Rate (CCR) of over 80% with the testing data of five gait-related papers [4]. Shahid et al. analyzed human gait patterns through vibrations from eight major joints using the Intelligent Gait Oscillation Detector (IGOD), and they found that each person had a different vibration intensity for each joint, resulting in different levels of usage for each joint when walking [5]. Shin and Jin used four infrared cameras and infrared-reflective markers to analyze movement patterns with walking speed, and they found that there were significant differences in the angle of the torso depending on walking speed [6].

For research identifying walking patterns through sensors, there is a concern that individuals will move differently than usual, due to the use of a large number of sensors or special wearable measuring equipment. In addition, by focusing on what is visually salient, the data obtained from image analysis could differ from the data derived using sensors. Specifically, data derivation methods using sensors and gait analysis methods on video footage differ in various aspects such as shooting angles, and experimental environments, etc. Therefore, there may be substantial differences between CCTV-based walking evidence images and sensor-based measurement data, which may reduce the validity of any collected walking characteristics. Most police officers use video data such as CCTV as evidence, so this problem greatly degrades the usability of analysis.

The continuously improving quality of video recorders has enabled detailed analysis of dynamic movements, and studies based on video data are underway. In a forensic gait analysis using image analysis of video footage, Zhang et al. proposed a method of classifying the age groups of young people and the elderly using a Hidden Markov Model (HMM) for the gait obtained by filming walking scenes three times per person for 14 subjects, and a CCR of over 70% was obtained [7]. Jang et al. conducted a gait analysis based on data on knee joints and glides obtained from smartphone photographs of the gaits of seven adult males [8].

Nevertheless, the existing image-based studies still have some limitations. First, the sample size must be large to achieve significant results [9], but the existing studies use significantly less data than the present study, thereby making them relatively less reliable. Second, there are fewer types of gait parameters observed. Although walking involves a variety of body parts, such as the neck and spine, existing studies have focused only on measuring lower extremity-related parameters. Third, the majority of experiments have been limited to lateral plane. Since certain parameters can only be visually checked from the front of the subjects, such as shoulder slope or knee varus/valgus when walking, it is necessary to measure the subject’s gait parameters not only from the lateral plane but also from the frontal plane.

The purpose of this study is to propose relevant key parameters for individual identification based on various parameters measured from a number of subjects. To this end, the gait patterns of 291 subjects were measured to complement the above studies and to increase their utility. In total, 34 parameters related to lower extremities, vertebrae, necks, etc. were measured in two planes (frontal and lateral), using four video cameras aiming toward four directions.

## 2. Materials and Methods

### 2.1. Gait Parameter

Table 1 lists the various gait parameters that have been used in past studies for gait analysis through image analysis of video footages. The gait parameters considered in this study include various elements used in previous gait analysis studies [10]. After reviewing the possibility of visual analysis of walking, final variables for measurement were selected (See Table 2 and Table 3). These were adopted because they were originally used for the same purposes as this study. The additional parameters of the Swing/Stance phase, Foot Valgus/Varus were also adopted based on the opinions of forensic gait analysis experts. Although Swing/Stance phase parameters do not show visual differences in images, they can be easily extracted computationally, and they have been used often in traditional gait analysis, so we adopt them [11,12,13,14,15]. Additionally, we added Foot Valgus/Varus because they can show a visually distinct difference in the images.

### 2.2. Experimental Environment

The data used in this study were obtained from four different cameras. The entire walkway is 10 m long, with force plates located in the middle. Each camera was placed three meters from the center of the pedestrian path, and the cameras at the front and back were about one meter from the end of the walkway. The walkway was firmly secured so that participants could walk comfortably without any separation or shaking.

### 2.3. Measurement Method of Gait Parameters

The data used in the analysis were extracted from the participants’ gait footages. The participants walked 10 times in a natural walking motion on the 6-m walkway seen in Figure 1. To determine the heel-strike and toe-off moments, we observed the sagittal plane video. To minimize the distortion of the camera lenses, we selected the image showing the participant in the middle of the walkway. The heel-strike moment is defined as when the angle of the foot is the largest, i.e., when the frontal foot heel-strikes. The toe-off moment is defined as when the angle of the foot is the largest when the backward foot toes off. The double support phase moment is defined as the midpoint between the heel-strike and toe-off moments. After determining these points, we measured each parameter with protractor software using an image of each moment. Each analysis was performed in a double check method in which one person analyzes and the other person checks the accuracy. Each parameter was measured based on anthropometric terms suggested by the previous study conducted by Jung et al. [16], which aimed to standardize anthropometric points. Table 2 and Table 3 show the definition and derivation method of each parameter.

### 2.4. Participants

We measured the gaits of 291 Korean adult males and females, most of them were not obese (the number of people who have a BMI between 30 and 35: 18 (6%)) and had no history of lower limb or spine injuries or illnesses. Participants were recruited from similar age groups and with similar gender distributions so that the results were not affected by age or gender (Table 4). Participants were fully informed of the study and each participant gave their consent to participate. The measurements were performed after being reviewed and approved by the University’s Institutional Review Board.

### 2.5. Statistical Analysis

In this study, the major variables that differ according to gender, age, and BMI are identified. K-means clustering analysis was used to divide the groups (large, medium, and small) of BMI. An Independent two-sample *t*-test was used in continuous scale variables for mean comparison between male and female groups. In category variables, statistical differences between groups were analyzed through cross-stabilization analysis and Chi-square test. The mean difference between the age group and the BMI group was analyzed using one-way ANOVA for continuous scale variables, and the difference between each group was identified using the Scheffé test. In addition, for categorical scale variables, cross-tabulation analysis and Chi-square test were used to determine differences between groups. All statistical analyses used SPSS software (v.23) (IBM, New York, NY, USA).

## 3. Results

### 3.1. Gait Characteristics and Key Parameters

The descriptive statistics for the measured gait parameters are listed in Table 5. To compare the variance of each parameter, a coefficient of variation (CV) was obtained; then, the value of the obtained coefficient of variation was multiplied by 100 for more convenient comparison. Overall, the coefficient of variation was measured high in the parameters associated with the spine and neck.

Next, K-means clustering analysis was performed to identify parameters with large coefficients of variation using three clusters (Table 6). Cluster 1 (Centroid = 69.59), which had the largest coefficient of variation, contained three gait parameters in total, Cluster 2 (Centroid = 39.00) contained five, and Cluster 3 (Centroid = 6.42) contained 22.

### 3.2. Differences by Age Groups

Next, independent two-sample *t*-tests of all parameters were conducted for the two age groups (young and elderly) using SPSS (ver.23). Participants in their 20s, 30s, and 40s were classified as the young group, while those in their 50s and 60s were classified as the elderly group. Among them, we examined parameters that could be considered meaningful with a significant probability of less than 0.05.

As a result, we found significant differences in lateral flexion of the spinal column at the double support phase, head movements in the sagittal plane at the heel-strike, head movements in the sagittal plane at the toe-off, and mean head movements in the sagittal plane (see Figure 2, Figure 3, Figure 4 and Figure 5). Furthermore, the parameters of lateral flexion of the spinal column at the double support phase, head movements in the sagittal plane at the heel-strike, head movements in the sagittal plane at the toe-off, and mean head movements in the sagittal plane were higher in the elderly group than in the young group. Meanwhile, no significant differences were found in left knee varus/valgus, right knee varus/valgus, foot varus/valgus, toe-in gait/toe-out gait, or pelvis abduction/adduction.

### 3.3. Difference by Gender

We performed independent two-sample *t*-tests on the gait parameters using gender as a factor. Figure 6, Figure 7, Figure 8, Figure 9, Figure 10, Figure 11, Figure 12, Figure 13, Figure 14, Figure 15, Figure 16, Figure 17, Figure 18, Figure 19, Figure 20 and Figure 21 show the results of significant gait parameters which have *p*-values less than 0.05. The following gait parameters had higher average values in women: the ratio of swing phase of left foot, lateral trunk flexion at the double support phase, forward/backward trunk bending at heel-strike, mean forward/backward trunk bending, head movements in the frontal plane at the double support phase, and head movements in the frontal plane at the double support phase. Meanwhile, the following gait parameters had higher average values in men: distance between the feet, step length, walking speed, foot angle at heel-strike, knee angle at heel-strike, flexion of the spinal column in the sagittal plane at heel-strike, flexion of the spinal column in the sagittal plane at toe-off, mean flexion of the spinal column in the sagittal plane, and shoulder angle in the frontal plane at the double support phase.

Cross-tabulation analysis was conducted on the nominal gait parameters, i.e., left knee varus/valgus, right knee varus/valgus, foot varus/valgus, toe-in gait/toe-out gait, and pelvis abduction/adduction. Among them, we only examined gait parameters that may be considered meaningful with *p*-values less than 0.05 (Table 7, Table 8 and Table 9).

Although significant gender-specific differences were found in the right knee varus/valgus, the contingency coefficient value was 0.16 and the relationship was not large. In both men and women, the neutral angle is more frequent than the right knee varus/valgus. In women, aside from neutrality, each of the right knee varus/valgus is distributed similarly. By contrast, in men, it is mostly distributed on the right knee valgus. In addition, the direction of the foot (toe-in gait/toe-out gait) varies significantly depending on gender. In men, the proportion of toe-out gait is quite high, while in women, the proportion of neutral walking is high. The abduction/adduction of pelvis also varies significantly depending on gender. In both men and women, the number of neutral walks is higher than either of pelvis abduction/adduction. However, pelvis adduction and pelvis abduction are distributed similarly in men, while women mostly show pelvis adduction.

### 3.4. Differences in Gait Characteristics by BMI

#### 3.4.1. Cluster Analysis by BMI

We conducted a K-means clustering analysis to divide the participants into three BMI groups. As a result, they were classified into Cluster 1 (mean = 23.94) as the middle group, Cluster 2 (mean = 29.45) as the large group, and Cluster 3 (mean = 19.68) as the small group (F = 584.84, *p*-value < 0.001).

#### 3.4.2. Differences by BMI Group

First, we conducted ANOVA to test the differences in gait parameters between BMI groups. The results showed that the following parameters had a significant impact on the gait of classified BMI groups: distance between the feet; the ratio of swing phase; the ratio of stance phase; foot angle at heel-strike; flexion of the spinal column in the sagittal plane at heel-strike; flexion of the spinal column in the sagittal plane at toe-off; mean flexion of the spinal column in the sagittal plane; lateral trunk flexion at the double support phase; lateral trunk flexion at the double support phase; head movements in the frontal plane at the double support phase; head movements in the sagittal plane at heel-strike; and mean head movements in the sagittal plane at heel-strike. Figure 22, Figure 23, Figure 24, Figure 25, Figure 26, Figure 27, Figure 28, Figure 29, Figure 30, Figure 31, Figure 32, Figure 33, Figure 34 and Figure 35 show the mean of each BMI group for each parameter, and the result of the post hoc test (Scheffé test) is denoted alphabetically. The same alphabet is interpreted as having the same population mean.

We also conducted a cross-analysis for nominal gait parameters such as toe-in/out and pelvis abduction/adduction. As a result, the level of BMI was found to have a significant impact on toe-in gait/toe-out gait as well as pelvis abduction/adduction (Table 10 and Table 11, respectively).

## 4. Discussion

### 4.1. Gait Parameter Difference between Individuals

Through the K-means clustering of the values of the coefficient of variation×100 for each parameter, gait parameters with significant differences between individuals could be identified. Based on the fact that the final cluster centroid values of Cluster 1 (Centroid = 69.59) and Cluster 2 (Centroid = 39.00) are high compared to that of Cluster 3 (Centroid = 6.42), the parameters belonging to Cluster 1 and Cluster 2 have significant individual differences compared to those belonging to Cluster 3. Specifically, we can confirm that most of the parameters belonging to Cluster 1 and Cluster 2 are for a particular part of the body, spine, and head. For cluster 1, we can see that all of these parameters involve flexion of the spinal column in the sagittal plane, and most of the parameters in Cluster 2 involve the head movements in the sagittal plane.

The largest individual difference was shown to be in the parameters related to flexion of the spinal column in the sagittal plane. This means that there are large individual differences in how much pedestrians lean forward when walking. Among the visually prominent variables in the results of Koreans’ walking characteristics, the main variable appears to be how far forward they bend and walk. The shapes of one’s neck and spine vary from individual to individual [17], and they depend on the size of one’s muscles [18]. This difference can cause differences between individuals in neck extension and spinal flexion when walking. The other relevant variables were the foot angle and foot spacing variables at the heel strike moment; these are all foot-related variables, meaning that there is a large individual difference in the angle and spacing of the toe when stepping on the ground. In particular, the foot-related variables have the advantage of being easily identifiable in images.

Furthermore, the knee varus or valgus is considered a significant parameter in identifying individuals. Of the total subjects, the knees of both legs were bent inward or outward from the body in 22 subjects, representing 7.5% of the total subjects, and those with both knees varus only represented 0.6% of the total. Similarly, toe-in gait is also considered a significant parameter in person identification. Only 1.7% of the total number of participants showed toe-in gait. Whether both knees are varus/valgus or toe-in gait are visible parameters that are only characteristic of a small proportion of people, we believe that they are important parameters in the identification of persons through images.

### 4.2. Key Parameters Related to Age Groups

As a result of conducting *t*-tests on all parameters to assess the difference between young and elderly people, the parameters of lateral flexion of the spinal column at the double support phase, head movements in the sagittal plane at heel-strike, head movements in the sagittal plane at toe-off, and mean head movements in the sagittal plane were found to differ significantly by age group. These results indicate that as people get older, their upper body shakes less to the left and right, and they walk more with their necks bent forward. According to Yoon et al., the Ground Reaction Force (GRF) of young people is much stronger than that of the elderly [19]. At this time, the stronger the repulsion force of the opposite foot of the preceding foot, the more the left-to-right bias increases, and it can be assumed that the result is such that the older the person, the less repulsion force they exert.

In general, the results also show that as people get older, their necks bend further forward as they walk. According to Kocur et al., muscle degeneration with aging causes FHP (front head posture), which increases linearly from the age of 28 [20]. Therefore, the degree of FHP in the elderly is severe, which may increase the range of the head movements in the sagittal plane while walking, thus increasing the angle of the head movements in the sagittal plane. In addition, due to muscle degeneration caused by aging, the higher the age group, the more likely one’s head is to lean forward, which can cause an increase in the range of head movements in the sagittal plane.

### 4.3. Key Parameters Related to Gender

The results of this study showed that, on average, men have different gait characteristics than women, such as a greater step length and a faster walking speed. In general, taller people have a relatively wider step distance [21]. The results of this study indicate that male participants are on average 11 cm taller than female participants. The walking speed results are also consistent with existing results showing that the average walking speed of men is higher than that of women [22]. In addition, it was found that men walk with their feet wider than women. Generally, when walking, women have a greater degree of hip internal rotation and adduction than men [23]. Hip internal rotation refers to the turning of the thigh bone inward in a hip joint (body side), while hip adduction is the position of the leg in the torso (inside). Therefore, the greater the degree of internal rotation and adduction of the hip joint when walking, the closer the leg is to the torso (inwards), so the distance between the feet tends to narrow. This explains the results showing that men have wider foot spacing than women when walking. The results showed that, for women, the percentage of time their feet are floating in the air is larger than that for men, and women also take steps faster than men. This can be seen from the results in this study indicating that the average swing phase value for women is greater than for men. Women may have a greater value of swing phase than men because they take narrower steps as well as more steps than men.

The results showed that, in men, the left shoulder is lower than the right shoulder when the left foot is in front of the right foot, and that the right shoulder is lower than the left shoulder when the right foot is in front at the double support phase. This means that there is a difference in the height of the left and right shoulders of men according to the advancing of the left foot and right foot at the double support phase. In other words, men tend to move their bodies from side to side and their shoulders up and down. It can also be observed that, compared to women, men walk with their heads tilted from side to side. Both men and women move their heads in the same direction depending on which foot is leading them when walking. However, men can be visually distinguished from women because of the large degree of frontal head movement. The results of these two parameters show that men tend to shake their bodies more from side to side when walking compared to women. These results are similar with the existing studies on the spinal, shoulder, and pelvic movements while walking [24]. Therefore, men can be visually distinguished from women because they have a relatively large degree of horizontal movement in walking.

According to the results, men generally have a greater knee angle when grounding compared to women. The greater the knee angle, the stronger the knee stretches. Therefore, men tend to walk with their knees straighter than women. This trend is consistent with existing studies showing that male knee extension is on average greater than that of females when grounding [25]. In addition, on ground grounding (heel-strike), the foot angle is generally greater than that of women. In other words, men have a greater degree of dorsiflexion at the point where the heel touches the ground than women. In general, ankle joint flexion of females is significantly smaller than that of males at the point of the ground during the walking cycle [22]. This is consistent with the results of this study, which show that the angle of feet of males on grounding is generally greater than that of females, as a larger flexion of the ankle joint corresponds to a greater angle of feet.

Overall, men walk with their upper bodies bent forward more than women. The results of this study show that men’s flexion of the spinal column in the sagittal plane when walking is higher than that in women. In addition, the results indicating that men have larger forward/backward bending values than women also show that the walking characteristics of men are distinct from those of women. The forward/backward trunk bending at toe-off does not show significant gender differences at the 0.05 levels in the t-test results. However, the average value for males is smaller than that for females, which can be interpreted to contribute to the same result as above. These walking properties, in which men bend their upper bodies further forward, are consistent with existing research results: According to Li et al., men’s cervical vertebrae are forward bent compared to women when walking [26]. In addition, Chung et al. confirmed that the male torso is tilted more forward than the female torso when walking [27].

The results showed that both men and women were more neutral than either of knee varus/valgus. With the exception of neutrality, men are mostly distributed in the knee valgus, while women are distributed similarly among knee varus/valgus. In other words, people who show knee valgus characteristics while walking are highly likely to be male. Meanwhile, men have an overwhelming number of cases of toe-out gait. In women, neutral gait is the most common. There are very few cases of toe-in gait in both men and women. Most men are a toe-out gait, and most women walk neutrally, so this parameter can be used to visually distinguish gender. In terms of pelvis abduction/adduction, both men and women are most likely to have neutral thighs that do not lean inward or outward from the body. Aside from neutrality, men have similar proportions of adduction and abduction, while women have a much larger proportion of adduction. In addition, compared to men, women tend to walk with their front legs moving inward. Previous studies have shown that hip internal rotation and adduction are greater in women than men, which explains why the walking characteristics of women are distinct from those of men [23].

### 4.4. Key Parameters Related to BMI

By analyzing the differences in gait parameters by BMI levels, we found several differences in gait characteristics among BMI groups. First, according to the study by Rosso et al., obese adults showed a larger distance between the feet [28]. We found a similar characteristic, that people in the large group walked with a larger distance between their feet compared to those in the small groups. This seems to be a habitual phenomenon in which the legs are spread out to walk comfortably depending on the thickness of the thighs. Rosso et al. also stated that obese adults have the gait characteristics of a shorter swing phase and a longer stance phase [28]. This study obtained the same results, where people in the larger BMI group showed a shorter swing phase and a longer stance phase than those in the small group. This means that the larger one’s BMI, the less time their feet stay in the air and the longer they stay on the ground; this seems to be the result of the greater weight. The small group showed a smaller heel-strike foot angle when walking compared to the large group. This means that the angle of the feet of people in the small group is smaller when they reach the ground than that of the large group. Considering that people in the small group show a long swing phase and a short stance phase, it can be said that the small group has less dynamic gait characteristics than the large group.

Flexion of the spinal column in the sagittal plane increases as BMI increases both at heel-strike and toe-off. These parameters demonstrate how much the spinal column bends when walking. This means that people in the large BMI group bend forward more while walking than those in the small group. Regarding the head movements with each foot forward in the frontal plane, the large group showed significantly more left and right movement of the head than other groups. The head movements in the sagittal plane tend to increase as BMI increases, which means that the neck is bent more while walking. In summary, a person with a high BMI tends to show more head movement in both the sagittal and frontal planes.

There were significant differences in the ratio of toe-in/toe-out gait between BMI groups. Small groups have a significantly higher proportion of neutral gait than other groups, as well as a significantly lower proportion of toe-out gait. By contrast, compared to the small group, the middle and large groups have a significantly higher proportion of toe-out gait. As a result, the smaller the body shape, the higher the percentage of front-facing toe steps, and the larger the body shape, the higher the percentage of walking with an outward-facing toe step. This result is consistent with the study of Hills and Parker, which stated that obese children walk with the toe facing outward [29]. This means that this study confirmed a close relationship between the body shape and the toe-out gait regardless of age.

### 4.5. Limitations

In this study, we did not collect dominant hand information from the participants. According to Spry et al., leg dominance in humans follows from the principal hand [30]. This means that if someone is right-handed, then they often prefer to use their right foot. In addition, Riskowski et al. stated that the existence of a symmetric or asymmetric gait in a healthy adult is determined by leg dominance [31]. According to the results of both studies, collecting information on the participants’ dominant hands is likely to help identify lower body movements while walking. Other limitations of this study are related to measurement errors. As written in Section 2.3, the data used in this study are manually extracted based on images. Therefore, measurement errors may exist in extracting the data. In future studies, it will also be meaningful to verify the accuracy and reliability of the variables proposed in this study through comparison between the visual variables analyzed in this study and the results analyzed by the 3D motion sensor.

## 5. Conclusions

In this study, we conducted ANOVA and k-mean clustering analysis on 34 parameters to suggest the gait characteristics by which individuals can be distinguished. We also utilized a large number of data compared to conventional gait studies to ensure the validity of the results. First, the most prominent parameters used for personal identification were those related to flexion of the spinal column and head movements in the sagittal plane. This means that the movement of the neck and upper body is a significant factor in identifying individuals. For the lower body, the foot angle and the distance between the feet showed significant differences while walking. Furthermore, it is considered a significant characteristic in identifying individuals in that fewer people show the characteristics of the knee varus/valgus. Second, the parameters that indicate differences by age group include lateral flexion of the spinal column, and overall head movements in the sagittal plane. Third, when men walk, compared to women, they bend their upper bodies more forward and shake their bodies more from side to side. In the case of the lower body, they stretch their knees and bend their ankles at heel-strike. They also have a short stance phase, a large distance between feet, a longer step length, and fast walking velocity. Finally, compared to women, men have a greater tendency to show the characteristics of the knee valgus and toe-out gait, and women have a higher rate of pelvis abduction. Finally, among the group with a large BMI, the upper body is bent further forward and sloped more to the left, and the neck is bent further forward and shows more lateral movements. Regarding the lower body, the stance phase is relatively longer for people in the large BMI group than it is for those in other groups because they take steps forward rather than upward steps, and the angle of the foot is greater when touching the ground. Finally, there is a high tendency to show the characteristics of the toe-out gait. The results of this study are expected to be effectively utilized to identify key parameters by which individuals can be distinguished through visual gait characteristics.

## Figures and Tables

**Figure 1 ijerph-19-02467-f001:**
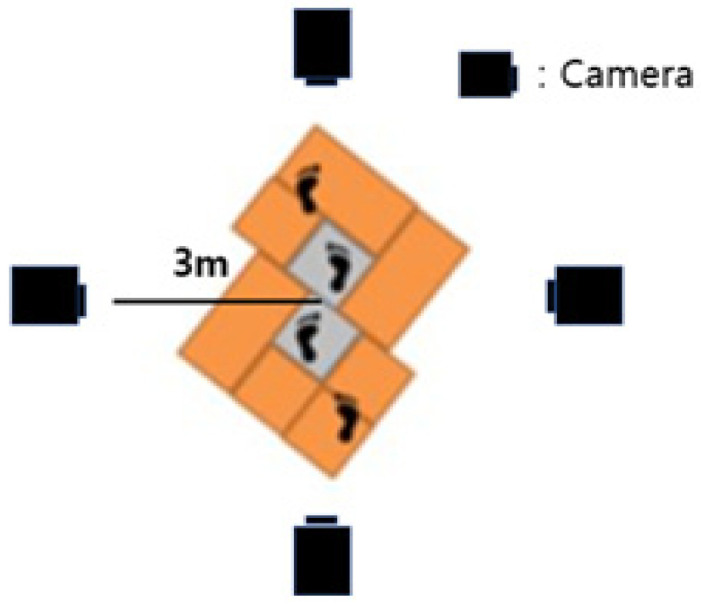
Experimental environment.

**Figure 2 ijerph-19-02467-f002:**
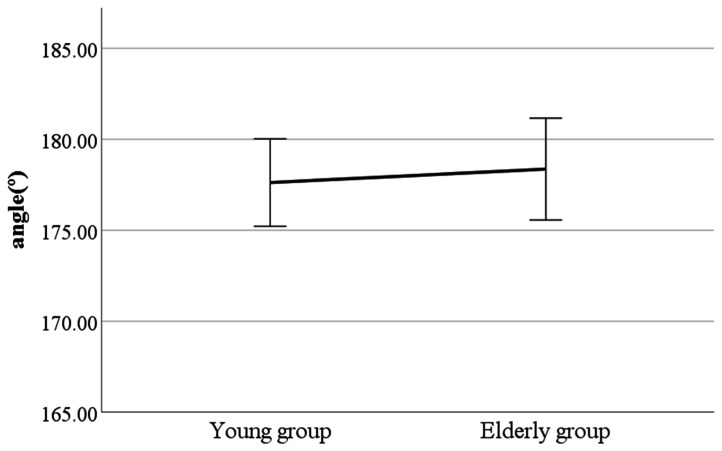
Age-to-age difference in double support phase (left foot forward) lateral flexion of the spinal column (F value: 2.31, DF: 289, *p*-value: 0.04).

**Figure 3 ijerph-19-02467-f003:**
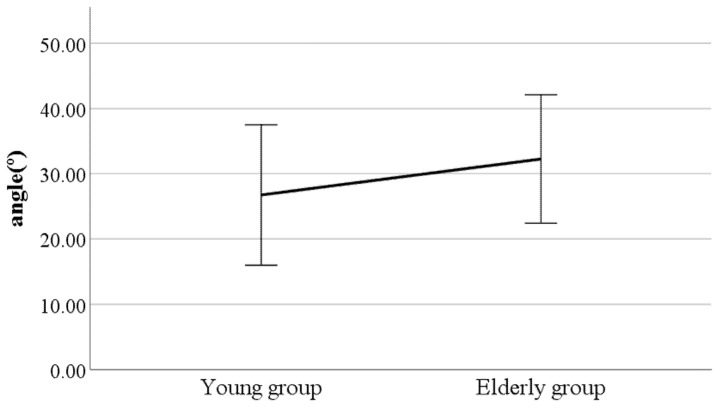
Age-to-age difference in heel-strike head movements in the sagittal plane (F value: 0.67, DF: 289, *p*-value: 0.00).

**Figure 4 ijerph-19-02467-f004:**
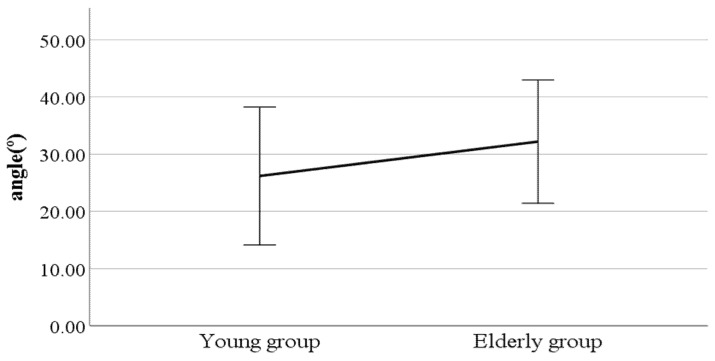
Age-to-age difference in toe-off head movements in the sagittal plane (F value: 3.04, DF: 289, *p*-value: 0.00).

**Figure 5 ijerph-19-02467-f005:**
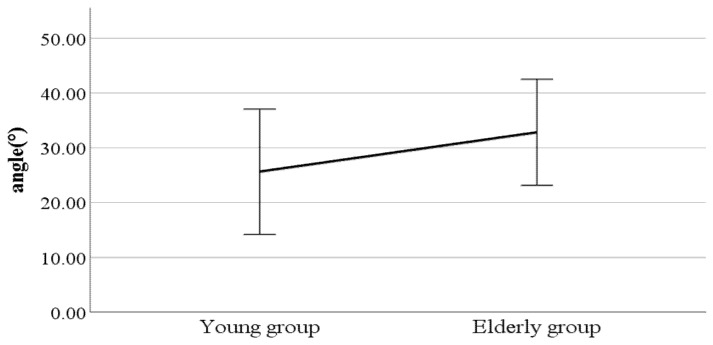
Age-to-age difference in mean head movements in the sagittal plane (F value: 2.88, DF: 289, *p*-value: 0.00).

**Figure 6 ijerph-19-02467-f006:**
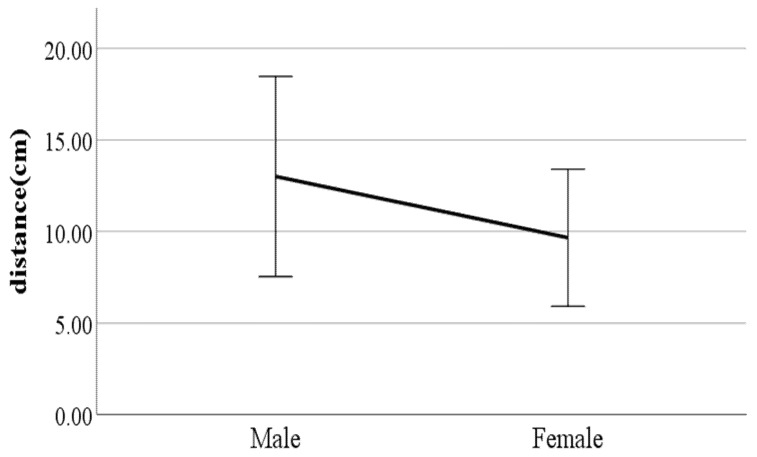
Gender difference in distance between the feet (F value: 13.40, DF: 289, *p*-value: 0.00).

**Figure 7 ijerph-19-02467-f007:**
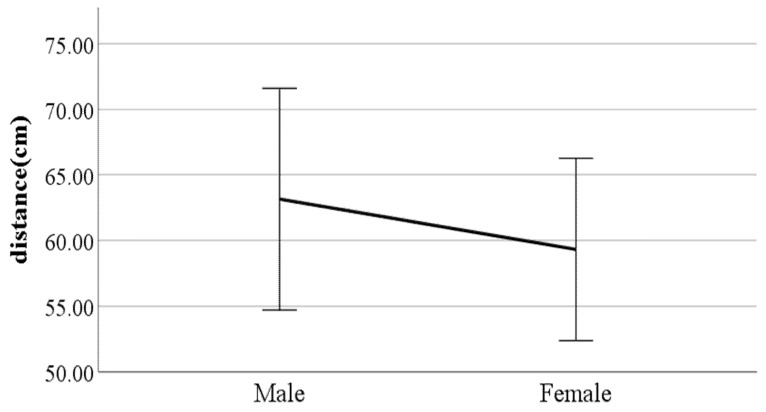
Gender difference in step length (F value: 5.78, DF: 289, *p*-value: 0.00).

**Figure 8 ijerph-19-02467-f008:**
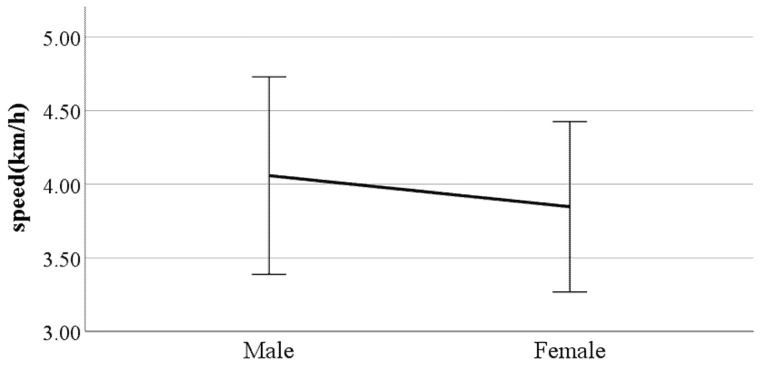
Gender difference in walking speed (F value: 3.58, DF: 289, *p*-value: 0.00).

**Figure 9 ijerph-19-02467-f009:**
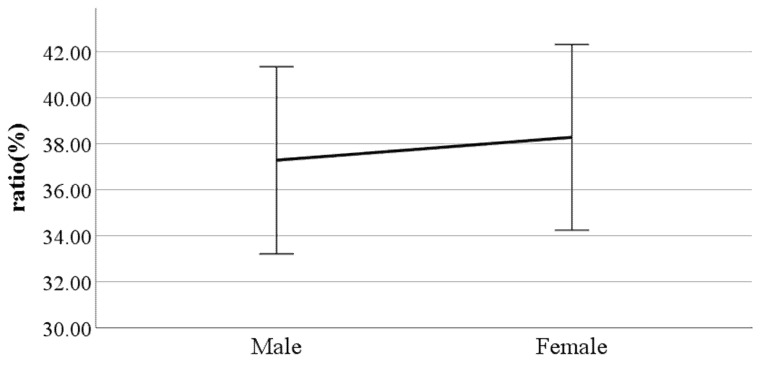
Gender difference in the ratio of swing phase (F value: 0.31, DF: 289, *p*-value: 0.04).

**Figure 10 ijerph-19-02467-f010:**
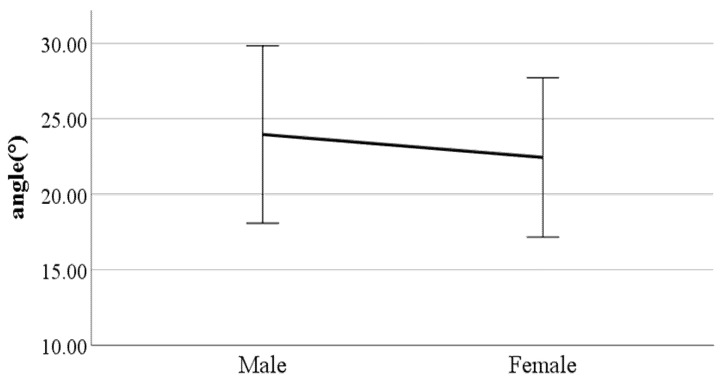
Gender difference in foot angle at heel-strike (F value: 0.82, DF: 289, *p*-value: 0.02).

**Figure 11 ijerph-19-02467-f011:**
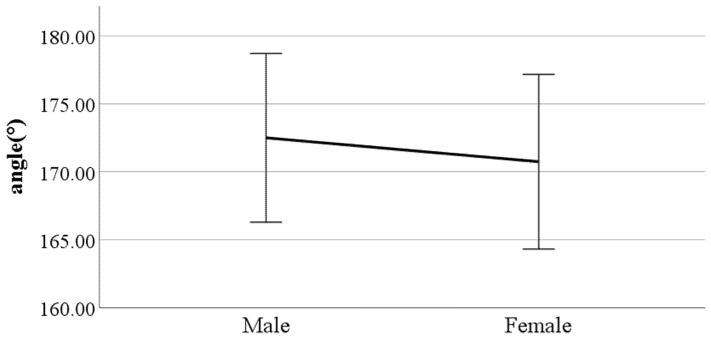
Gender difference in knee angle at heel-strike (F value: 2.39, DF: 289, *p*-value: 0.02).

**Figure 12 ijerph-19-02467-f012:**
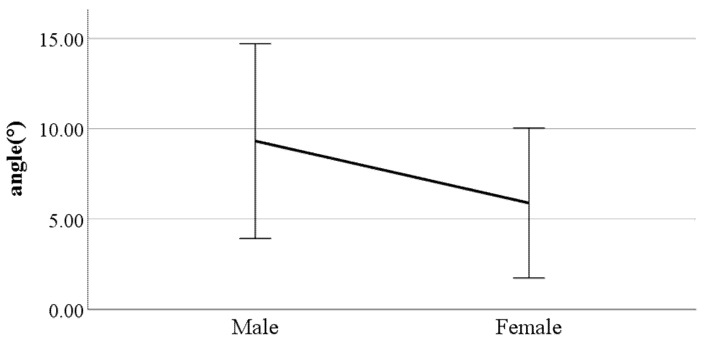
Gender difference in the flexion of the spinal column in the sagittal plane at heel-strike (F value: 11.29, DF: 289, *p*-value: 0.00).

**Figure 13 ijerph-19-02467-f013:**
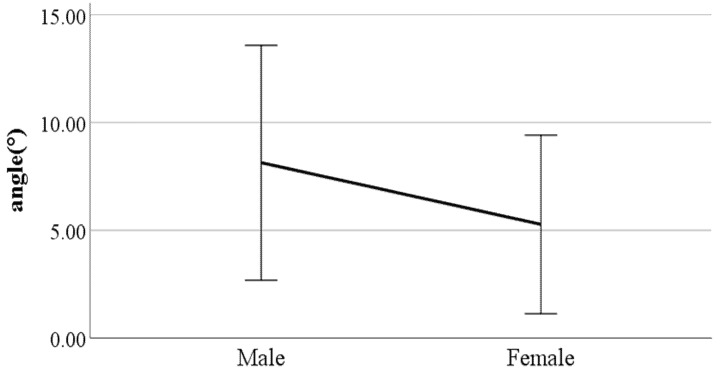
Gender difference in the flexion of the spinal column in the sagittal plane at toe-off (F value: 17.11, DF: 289, *p*-value: 0.00).

**Figure 14 ijerph-19-02467-f014:**
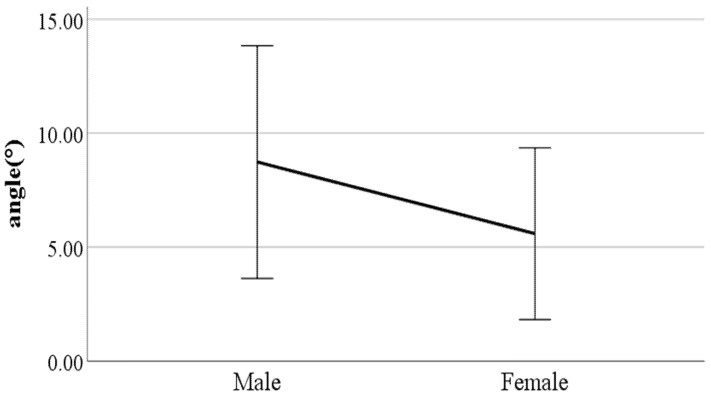
Gender difference in mean flexion of the spinal column in the sagittal plane (F value: 24.67, DF: 289, *p*-value: 0.00).

**Figure 15 ijerph-19-02467-f015:**
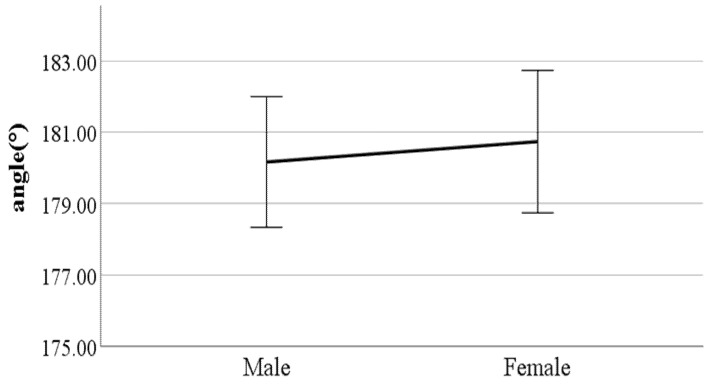
Gender difference in lateral trunk flexion at the double support phase (left foot forward (F value: 0.83, DF: 289, *p*-value: 0.01).

**Figure 16 ijerph-19-02467-f016:**
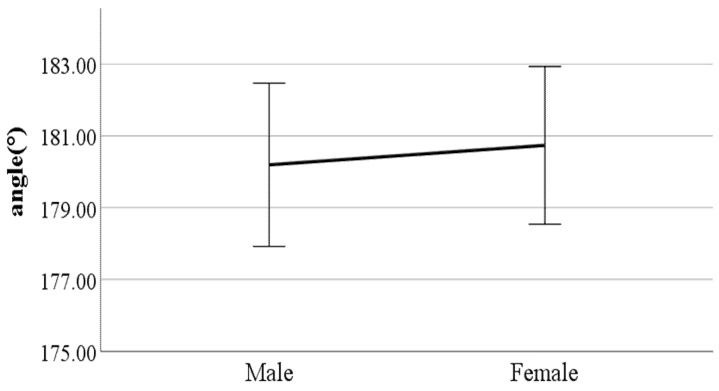
Gender difference in lateral trunk flexion at the double support phase (right foot forward) (F value: 0.01, DF: 289, *p*-value: 0.04).

**Figure 17 ijerph-19-02467-f017:**
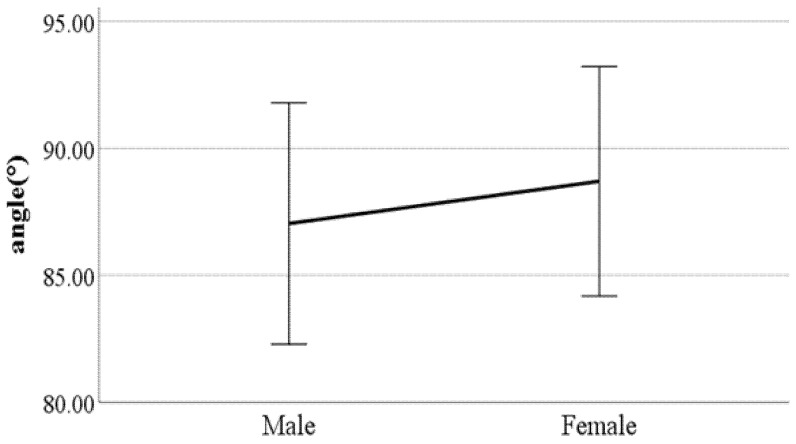
Gender difference in forward, backward trunk bending at heel-strike (F value: 0.19, DF: 289, *p*-value: 0.00).

**Figure 18 ijerph-19-02467-f018:**
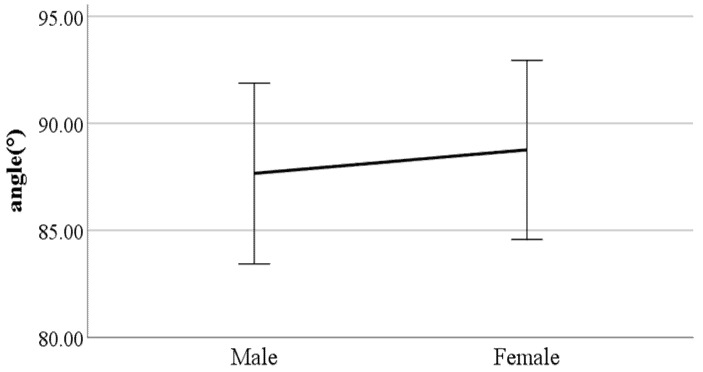
Gender difference in mean forward/backward trunk bending (F value: 0.22, DF: 289, *p*-value: 0.03).

**Figure 19 ijerph-19-02467-f019:**
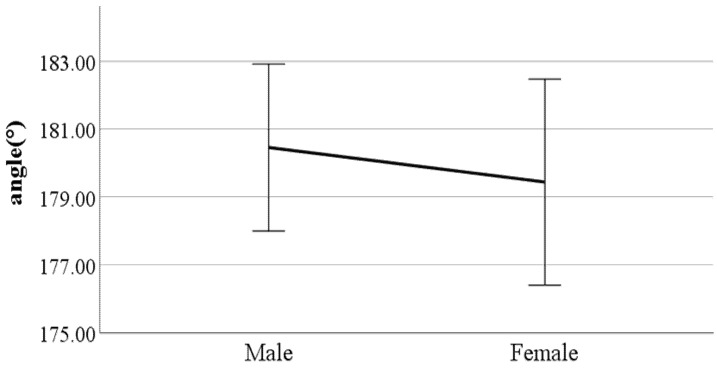
Gender difference in shoulder angle in the frontal plane at the double support phase (left foot forward) (F value: 4.72, DF: 289, *p*-value: 0.00).

**Figure 20 ijerph-19-02467-f020:**
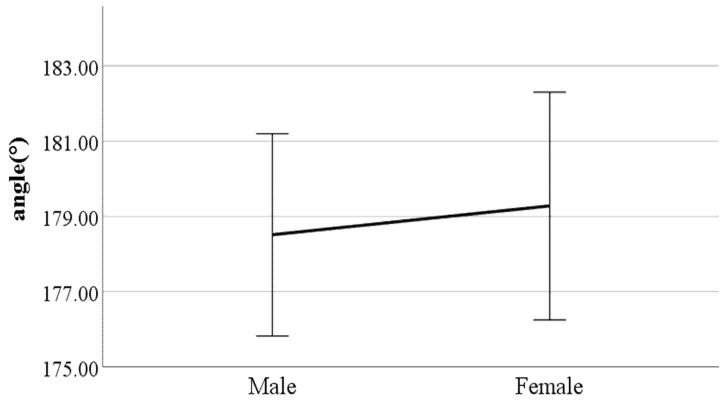
Gender difference in head movements in the frontal plane at the double support phase (left foot forward) (F value: 3.37, DF: 289, *p*-value: 0.02).

**Figure 21 ijerph-19-02467-f021:**
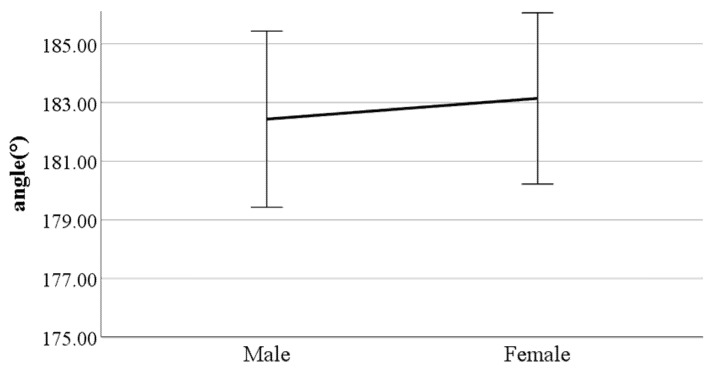
Gender difference in head movements in the frontal plane at the double support phase (right foot forward) (F value: 0.03, DF: 289, *p*-value: 0.04).

**Figure 22 ijerph-19-02467-f022:**
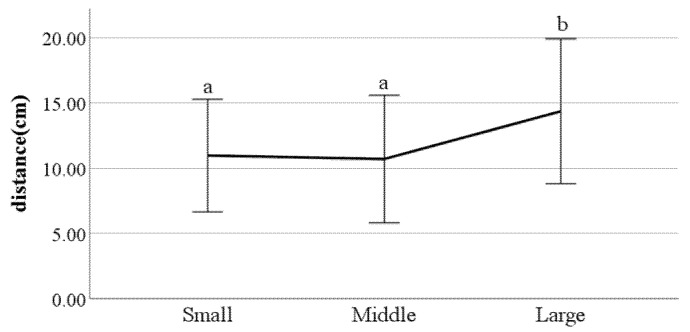
Distance between the feet (F value: 0.08, DF: 2, *p*-value: 0.00).

**Figure 23 ijerph-19-02467-f023:**
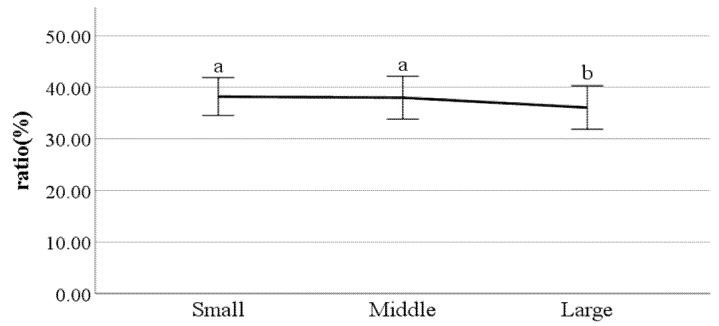
Swing phase of left foot (F value: 4.56, DF: 2, *p*-value: 0.01).

**Figure 24 ijerph-19-02467-f024:**
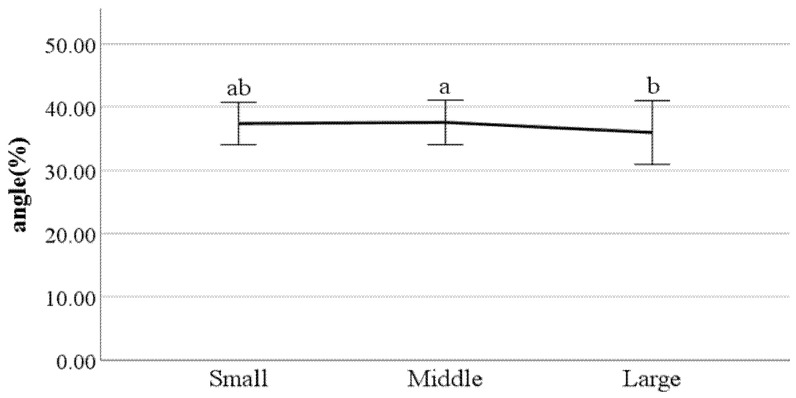
The ratio of swing phase of right foot (F value: 3.17, DF: 2, *p*-value: 0.04).

**Figure 25 ijerph-19-02467-f025:**
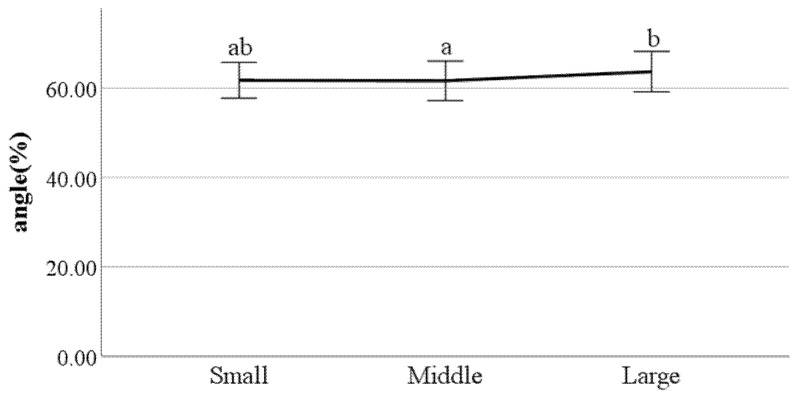
The ratio of stance phase of left foot (F value: 3.93, DF: 2, *p*-value: 0.02).

**Figure 26 ijerph-19-02467-f026:**
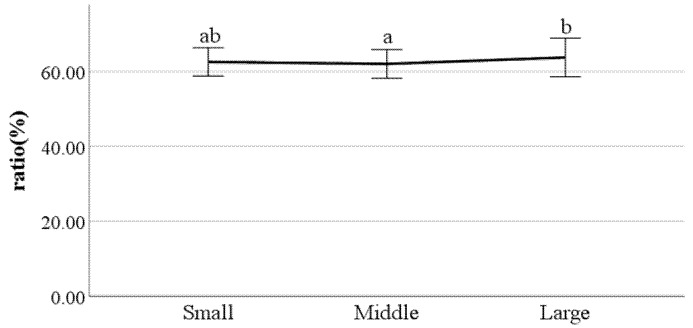
The ratio of stance phase of right foot (F value: 3.12, DF: 2, *p*-value: 0.04).

**Figure 27 ijerph-19-02467-f027:**
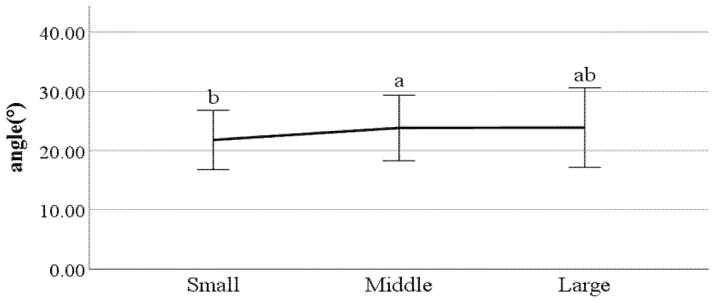
Foot angle at heel-strike (F value: 4.19, DF: 2, *p*-value: 0.02).

**Figure 28 ijerph-19-02467-f028:**
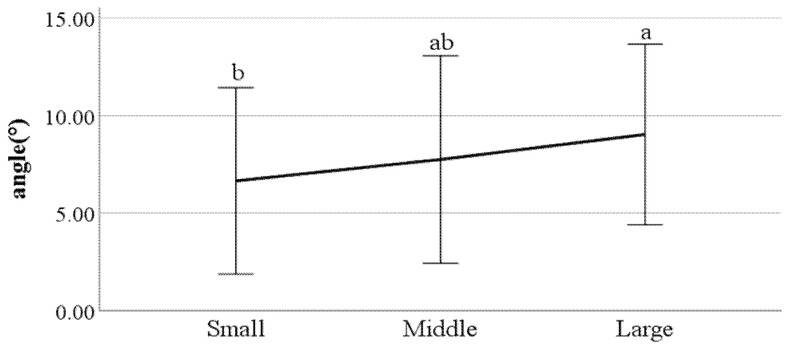
Flexion of the spinal column in the sagittal plane at heel-strike f (F value: 3.39, DF: 2, *p*-value: 0.04).

**Figure 29 ijerph-19-02467-f029:**
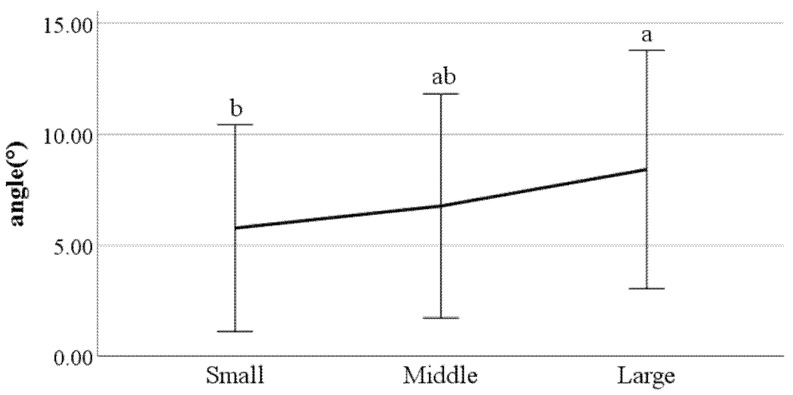
Flexion of the spinal column in the sagittal plane at toe-off (F value: 4.14, DF: 2, *p*-value: 0.02).

**Figure 30 ijerph-19-02467-f030:**
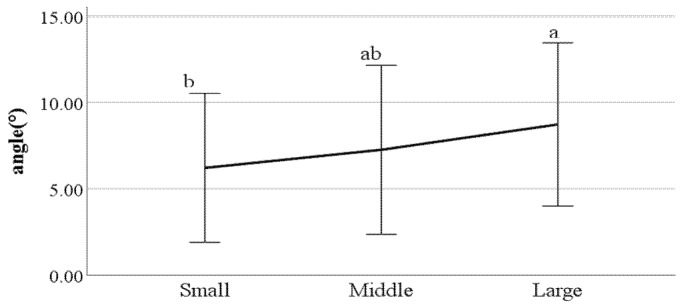
Mean flexion of the spinal column in the sagittal plane (F value: 4.29, DF: 2, *p*-value: 0.02).

**Figure 31 ijerph-19-02467-f031:**
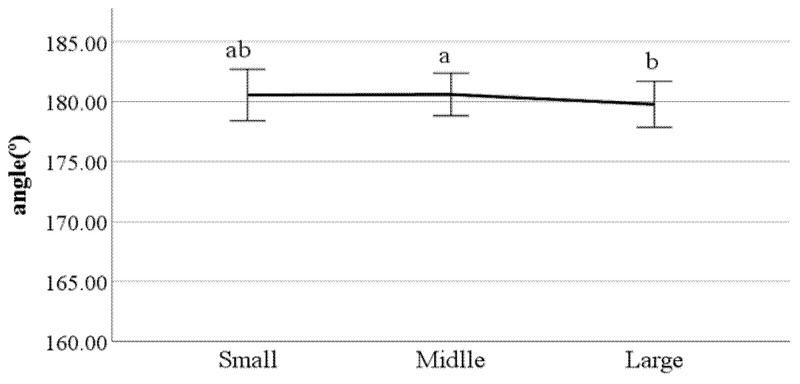
Lateral trunk flexion at the double support phase (left foot forward) (F value: 3.19, DF: 2, *p*-value: 0.04).

**Figure 32 ijerph-19-02467-f032:**
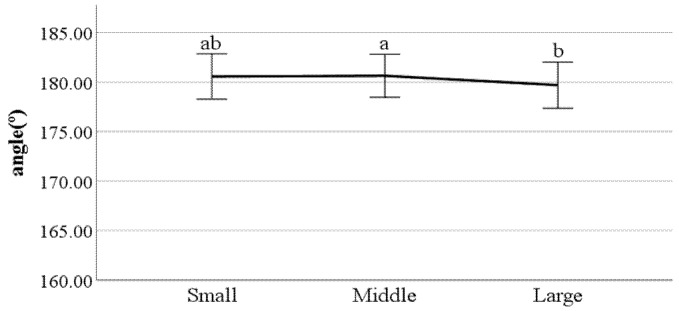
Lateral trunk flexion at the double support phase (right foot forward) (F value: 3.10, DF: 2, *p*-value: 0.04).

**Figure 33 ijerph-19-02467-f033:**
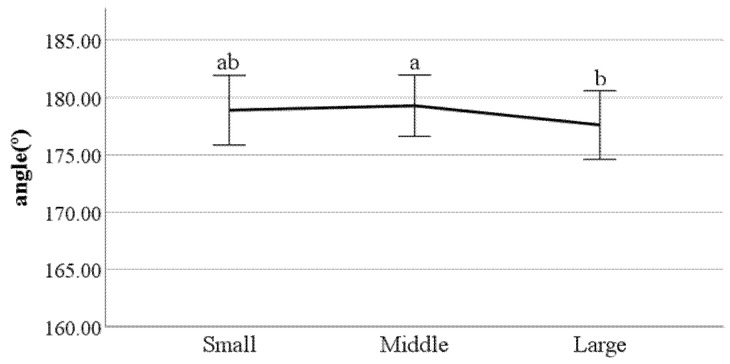
Head movements in the frontal plane at the double support phase (left foot forward) (F value: 5.90, DF: 2, *p*-value: 0.00).

**Figure 34 ijerph-19-02467-f034:**
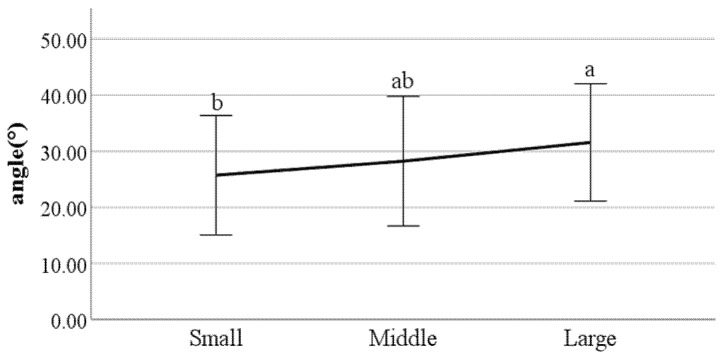
Head movements in the sagittal plane at heel-strike (F value: 4.13, DF: 2, *p*-value: 0.02).

**Figure 35 ijerph-19-02467-f035:**
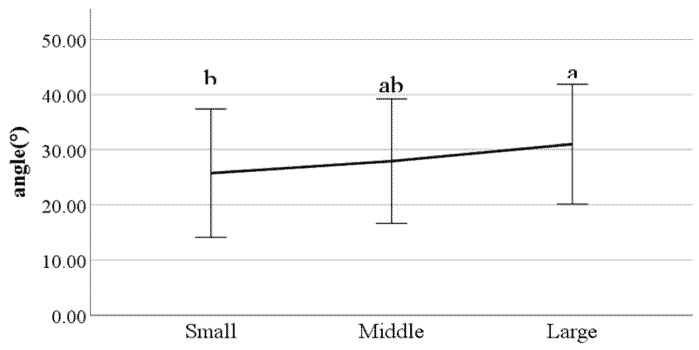
Mean head movements in the sagittal plane at heel-strike (F value: 3.21, DF: 2, *p*-value: 0.04).

**Table 1 ijerph-19-02467-t001:** Major gait parameters previously studied.

Gait Parameters	Previous Studies
Distance between the feet	Larsen, Simonsen, and Lynnerup, 2008 [10], Birch, Vernon, Walker, and Young, 2015 [14]
Step length	Birch, Vernon, Walker, and Young, 2015 [14], Prakash, Kumar, and Mittal, 2016 [11]
Number of steps per minute	Birch, Vernon, Walker, and Young, 2015 [14], Prakash, Kumar, and Mittal, 2016 [11]
Walking speed	Birch, Vernon, Walker, and Young, 2015 [14], Prakash, Kumar, and Mittal, 2016 [11]
Swing/Stance phase	Prakash, Kumar, and Mittal, 2016 [11]
Foot angle	Larsen, Simonsen, and Lynnerup, 2008 [10]
Knee angle	Larsen, Simonsen, and Lynnerup, 2008 [10]
Flexion of the spinal column in the sagittal plane	Larsen, Simonsen, and Lynnerup, 2008 [10]
Forward/Backward bending of upper body	Larsen, Simonsen, and Lynnerup, 2008 [10]
Rotation of the upper body	Larsen, Simonsen, and Lynnerup, 2008 [10]
Shoulder angle in the frontal plane	Larsen, Simonsen, and Lynnerup, 2008 [10]
Head movements in the sagittal plane	Larsen, Simonsen, and Lynnerup, 2008 [10]
Head movements in the frontal plane	Larsen, Simonsen, and Lynnerup, 2008 [10]
Toe-in gait/Toe-out gait	Birch, Vernon, Walker, and Young, 2015 [14]
Pelvis Abduction/Adduction	Larsen, Simonsen, and Lynnerup, 2008 [10]
Knee Varus⁄Valgus	Larsen, Simonsen, and Lynnerup, 2008 [10]

**Table 2 ijerph-19-02467-t002:** Definition and derivation method of gait parameters of the lower body.

Gait Parameters	Definition and Derivation Method
Distance between the feet	Definition: The distance between the centers of both feet.Derivation method: Derive from the captured image of the frontal plane at the double support moment. Calculate the horizontal distance of two feet based on the ratio of the image and the actual distance.
Step length	Definition: Distance between end-points of the front foot and back foot.Derivation method: Derive from the captured image of the sagittal plane at the toe-off moment. Draw a line between the end-points of the front foot and the back foot, then measure it in consideration of the ratio of the image and the actual distance.
Number of steps per minute	Definition: Number of steps per minute.Derivation Method: Calculate the number of steps taken in the total walk and divide it by the time taken.
Walking speed	Definition: Distance traveled per hour.Derivation method: Calculate the ratio of total walking distance to time.
The ratio of swing phase of left/right foot	Definition: The ratio of feet away from the ground to an entire step.Derivation method: Find the percentage of walking time that the corresponding foot is floating.
The ratio of stance phase of left/right foot	Definition: Percentage of time foot is touching the ground.Derivation method: Find the percentage of time that the corresponding foot supports the body.
Foot angle at heel-strike	Definition: The largest angle of the tip of the foot at heel-strike.Derivation method: Derive from a sagittal image of the heel-strike when the tiptoe is lifted most. Draw a line between the heel and the tiptoe, then measure the angle between this line and the ground.
Foot angle at toe-off	Definition: The largest angle of the tip of the foot at toe-off.Derivation method: Derive from a sagittal image of toe-off when the heel is lifted most. Draw a line between the heel and the tiptoe, then measure the angle between this line and the ground.
Knee angle at heel-strike	Definition: Knee angle at heel-strike.Derivation method: Derive from a sagittal image of heel-strike when the tiptoe is lifted most. Draw a line between the pelvis-knee-ankle, then measure the angle between these two lines.
Knee angle at toe-off	Definition: Knee angle at toe-off.Derivation method: Derive from a sagittal image of toe-off when the heel is lifted most. Draw a line between the pelvis-knee-ankle, then measure the angle between the two lines.
Knee Valgus/Varus	Definition: The pattern of the knee facing inside (valgus) or outside (varus) of the body.Derivation method: Derive from a frontal image of an individual standing with both feet before starting walking. Draw a line between the pelvis-knee-ankle, then measure the exterior angle between the two lines. Considering the measuring error, the values are divided into ≥90 as varus, <170 as valgus, and in between as neutral.
Foot Valgus/Varus	Definition: The pattern of the inside of the foot touching first (valgus) or the outside of the foot touching first (varus) at heel-strike.Derivation method: Derive from a frontal image when standing with both feet before starting to walk. Divide into valgus those with larger than 1°of valgus, varus those with larger than 3° of varus and others into neutral.
Toe-in gait/Toe-out gait	Definition: The direction of the end of toe at heel-strike.Derivation method: Derive from a frontal image at heel-strike. Divide into toe-in gait if the tiptoe faces the inside of the body, and toe-out gait if the tiptoe faces the outside of the body, and the rest as neutral.

**Table 3 ijerph-19-02467-t003:** Definition and derivation method of parameters of the upper body.

Name of Parameter	Definition and Derivation Method
Pelvis Abduction/Adduction	Definition: The pattern of the thigh moving inward (adduction) or outward (abduction).Derivation method: Derive from a frontal image at the double support phase (mid-point of heel-strike and toe-off). Pelvis adduction is when the front foot crosses or overlaps the end-point of the back foot toward the inside of the body in the transverse plane, and pelvis abduction is when the front foot is facing the outside of the body and the knee and toe are placed outside the pelvis. The remaining cases are classified as neutral.
Lateral flexion of the spinal column at the double support phase (left/right foot forward)	Definition: Lateral flexion of the spinal column while left or right foot remains forward.Derivation method: Derive from a frontal image of double support phase while left or right foot remains frontal. Draw two lines between the Sellion, Anterior Neck, and Anterior Waist, then measure the angle between the two lines; <180 means left tilt of the spinal column while >180 means right tilt of the spinal column.
Flexion of the spinal column in the sagittal plane at heel-strike/toe-off	Definition: Flexion of spinal column in the sagittal plane at heel-strike /toe-off.Derivation method: Derived from a sagittal image of heel-strike/toe-off. Draw a line between the Lateral Neck and the Lateral Waist, then measure the angle of the line from vertical.
Lateral trunk flexion at the double support phase (left/right foot forward)	Definition: The overall lateral flexion of the trunk in the frontal plane. (Left or right foot forward).Derivation method: Derive from the captured image of the frontal plane at the double support phase (left or right foot remaining forward). Draw a line between the Anterior Neck and the Anterior Waist and measure the angle of the line from vertical; <180 means left tilted body and >180 means right tilted body.
Forward/backward trunk bending at heel-strike/toe-off	Definition: The overall bending of the trunk in the sagittal plane at heel-strike, and at toe-off.Derivation method: Derive from the captured image of the sagittal plane at heel-strike (toe-off). Draw a line between the Lateral Neck-pelvis and measure the angle of the line from the horizon.
Shoulder angle in the frontal plane at the double support phase (left/right foot forward)	Definition: Angle due to difference in shoulder height when left or right foot is forward.Derivation method: Derive from a frontal image of a double support phase when the left or right foot remains frontal. Draw a line between the right and left Lateral Shoulders, then measure the angle of the line from the horizon. Classify as less than 180 degrees if the left shoulder is higher than the right shoulder horizontal and greater than 180 degrees if the left shoulder is lower.
Head movements in the frontal plane at the double support phase (left/right foot forward)	Definition: Overall head movements in the frontal plane during the double support phase (left, right foot forward).Derivation method: Derive from a frontal image of a double support phase when the left or right foot is forward. Draw a line between the Anterior Neck-Sellion, then measure the angle of the line from vertical.
Head movements in the sagittal plane at heel-strike, toe-off	Definition: Overall head movements in the sagittal plane during the double support phase (left, right foot forward).Derivation method: Derive from a sagittal image of heel-strike and toe-off. Draw a line between the Cervical-Anterior Neck, then measure the angle of the line from vertical.

**Table 4 ijerph-19-02467-t004:** The information of participants.

	Age					Total	Height	Weight	BMI
	20s	30s	40s	50s	60s				
Male	45	32	27	25	16	145	173 (1.1)	73.4 (9.8)	24.2 (2.8)
Female	46	30	25	29	16	146	162.9 (4.5)	57.5 (9.6)	21.6 (.3.1)
Total	91	62	52	54	32	291	170.6 (7.3)	68.7 (12.1)	23.5 (3.1)

**Table 5 ijerph-19-02467-t005:** Descriptive statistics for each parameter.

	N	MIN	MAX	MEAN	STD	VAR	CV*100
Distance between the feet (cm)	291	0.00	29.34	11.32	4.97	24.70	43.89
Step length (cm)	291	45.05	79.90	61.23	7.96	63.30	12.99
Number of steps per second	291	1.20	2.50	1.79	0.18	0.03	9.95
Number of steps per minute	291	72.07	150.00	107.16	10.68	114.00	9.96
Walking velocity (km/h)	291	2.65	5.87	3.95	0.63	0.40	16.04
The ratio of swing phase of left foot (%)	291	25.53	52.59	37.77	4.07	16.60	10.79
The ratio swing phase of right foot (%)	291	22.75	50.92	37.29	3.76	14.14	10.08
The ratio of stance phase of left foot (%)	291	47.41	74.47	61.94	4.35	18.91	7.02
The ratio of stance phase of right foot (%)	291	49.08	77.25	62.42	4.07	16.57	6.52
Foot angle at heel-strike (°)	291	8.20	39.99	23.19	5.63	31.67	24.26
Foot angle at toe-off (°)	291	26.24	78.26	56.41	10.61	112.66	18.82
Knee flexion at heel-strike (°)	291	141.51	180.00	171.62	6.37	40.52	3.71
Knee flexion at toe-off (°)	291	113.57	176.22	138.92	11.53	133.00	8.30
Lateral flexion of the spinal column at the double support phase (left foot forward) (°)	291	170.80	184.50	177.73	2.56	6.55	1.44
Lateral flexion of the spinal column at the double support phase (right foot forward) (°)	291	177.00	190.90	183.84	2.76	7.61	1.50
Flexion of the spinal column in the sagittal plane at heel-strike (°)	291	0.00	24.12	7.59	5.10	25.97	67.12
Flexion of the spinal column in the sagittal plane at toe-off (°)	291	0.00	23.20	6.70	5.04	25.41	75.28
Mean flexion of the spinal column in the sagittal plane (°)	291	0.00	21.70	7.15	4.75	22.56	66.38
Lateral trunk flexion at the double support phase (left foot forward) (°)	291	173.60	185.00	180.45	1.94	3.76	1.07
Lateral trunk flexion at the double support phase (right foot forward) (°)	291	174.30	186.10	180.47	2.25	5.05	1.25
Forward⁄backward trunk bending at heel-strike (°)	291	75.37	101.83	87.87	4.70	22.12	5.35
Forward⁄backward trunk bending at toe-off (°)	291	72.58	101.93	88.54	4.65	21.64	5.25
Mean Forward⁄backward leaning (°)	291	73.98	100.89	88.21	4.23	17.93	4.80
Shoulder angle in the frontal plane at the double support phase (left foot forward) (°)	291	173.20	187.30	179.95	2.81	7.89	1.56
Shoulder angle in the frontal plane at the double support phase (right foot forward) (°)	291	170.10	186.00	177.42	2.90	8.44	1.64
Head movements in the frontal plane at the double support phase (left foot forward) (°)	291	170.00	186.00	178.90	2.89	8.33	1.61
Head movements in the frontal plane at the double support phase (right foot forward) (°)	291	173.40	189.70	182.79	2.98	8.87	1.63
Head movements in the sagittal plane at heel-strike (°)	291	1.45	53.23	27.94	11.25	126.48	40.26
Head movements in the sagittal plane at toe-off (°)	291	0.32	58.57	27.49	12.46	155.33	45.34
Mean head movements in the sagittal plane (°)	291	1.80	55.09	27.71	11.42	130.44	41.21

**Table 6 ijerph-19-02467-t006:** The results of K-means clustering analysis of the CVs of gait parameters.

Groups	Gait Parameters
The coefficient of variation—Large group	Heel-strike flexion of the spinal column in the sagittal plane; toe-off flexion of the spinal column in the sagittal plane; and mean flexion of the spinal column in the sagittal plane
The coefficient of variation—Middle group	Toe-off head movements in the sagittal plane; distance between the feet; mean head movements in the sagittal plane; heel-strike head movements in the sagittal plane; and heel-strike foot angle
The coefficient of variation—Small group	Toe-off foot angle; walking velocity; step length; left foot swing phase right foot swing phase; number of steps per minute; number of steps per second; toe-off knee flexion; right foot stance phase; left foot stance phase; heel-strike, toe-off forward⁄backward trunk bending; heel-strike, toe-off forward⁄backward trunk bending; mean forward⁄backward leaning; heel-strike knee flexion; double support phase (right foot forward) shoulder angle in the frontal plane; double support phase (right foot forward) head movements in the frontal plane; double support phase (left foot forward) head movements in the frontal plane; double support phase (left foot forward) shoulder angle in the frontal plane; double support phase (right foot forward) lateral flexion of the spinal column; double support phase (left foot forward) lateral flexion of the spinal column; double support phase (left foot forward) lateral trunk flexion; and double support phase (right foot forward) lateral trunk flexion

**Table 7 ijerph-19-02467-t007:** Cross tabulation analysis of right knee varus/valgus by gender (X² = 7.83, DF = 2, *p*-value = 0.02).

	Knee Varus	Neutral	Knee Valgus	Total
Male	3	126	16	145
Female	14	120	12	146
Total	17	246	28	291

**Table 8 ijerph-19-02467-t008:** Cross tabulation analysis of toe-in gait/toe-out gait by gender (X² = 52.29, DF = 2, *p*-value = 0.00).

	Toe-In Gait	Neutral	Toe-Out Gait	Total
Male	1	37	107	145
Female	4	96	46	146
Total	5	133	153	291

**Table 9 ijerph-19-02467-t009:** Cross tabulation analysis of pelvis adduction/abduction by gender (X² = 19.41, DF = 2, *p*-value = 0.00).

	Pelvis Adduction	Neutral	Pelvis Abduction	Total
Male	34	86	25	145
Female	53	89	4	146
Total	87	175	29	291

**Table 10 ijerph-19-02467-t010:** Toe-in gait/toe-out gait (X² = 18.91, df = 4, *p*-value = 0.00).

	Toe-In Gait	Neutral	Toe-Out Gait	Total
Middle	2	61	94	157
Large	1	14	28	43
Small	2	58	31	91
total	5	133	153	291

**Table 11 ijerph-19-02467-t011:** Pelvis abduction/adduction (X² = 10.75, df = 4, *p*-value = 0.03).

	Pelvis Abduction	Neutral	Pelvis Adduction	Total
Middle	54	82	21	157
Large	11	28	4	43
Small	22	65	4	91
total	87	175	29	291

## Data Availability

The data that support the findings of this study are available from the corresponding author, upon reasonable request.

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
