# Peer review of "Identification of the Visually Prominent Gait Parameters for Forensic Gait Analysis"

_ijerph, 2022, doi:10.3390/ijerph19042467_

Round 1
Reviewer 1 Report
The study “Identification of the Visually Prominent Gait Parameters for Forensic Gait Analysis” aims to determine parameters to help identification of people via video analysis. Beyond all doubt this is a highly interesting approach, not only in order to identify people via e.g. CCTV but more interestingly in regard to underlying diseases ( e.g. hip osteoarthritis) – a possible application would also be interesting in regard to changed walking patterns due to hip/knee osteoarthritis. Thereby the walking patterns might be interesting to detect patients/people with combined diseases as mentioned above, as a medical application.
Although this is an innovative approach and seems promising for medical applications as well, there are some aspects, that should be addressed.
- Page 1, line 10: please change “walking” to "walking patterns”
- Page 1, line 20-23 please rephrase
- Page 2, line 90-97 please consider reformatting e.g.
- Distance between the feet
- Step length
- Number of steps per minute
- Page 3, line: adapted not adopted – please rephrase, sentence is not comprehensive
- Page 3, line 102: additionally not finally
- A major drawback is the limited information on participants – do they have any underlying musculoskeletal diseases (osteoarthristis, sport injuries etc.), please give demographic information on the patients in a table age, sex, osteoarthritis of the hip etc. and arrange the groups accordingly; a more distinct description of the sample is essential
- The aspects as mentioned above are critical!!!
- Please consider taking advantage of options such as support by an English native speaker to improve the language used in this manuscript
- It remains unclear whether one or more observers were included???
Overall this is a highly interesting study, yet there are some major drawbacks as described above, however there is a huge potential in this study.
Author Response
Thank you very much for the comments. The responses are in the attached file, please see the attachment.

Reviewer 2 Report
Dear Authors
Thank you for good opportunity.
There were some things in this article that I evaluated, such as the number of subjects. However, the following are concerns.
・It is unclear whether the purpose of the research is related to forensic science. The study that author referred to analyzed the gait of the assailant recorded in the surveillance camera images, but this study was of a healthy person, and it seems to be just a summary of the results of two-dimensional gait analysis data. We need to focus on the issues a little more. For example, the current status and problems of Gait Analysis in Forensic Medicine need to be clearly stated. There is also not enough information on what problems are caused by the small number of people in previous studies.
・Not enough information on ethical considerations.
・No information on statistical methods is provided.
・BMI is used as a key parameter, but information on indicators such as BMI, height, and weight are not given.
・In the introduction, it is written that research in two dimensions is beneficial, but within the limitations, analysis in three dimensions is necessary. If there is such a discrepancy in the description, it is unclear why this study was conducted in two dimensions.
・The content of the discussion is mostly results, with little or no discussion.
Author Response

(The authors gave the same response as above.)

Round 2
Reviewer 1 Report
I would like to congratulate the authors on their obvious effort to improve the manuscript. The overall quality has been greatly improved, the afore mentioned drawbacks have been addressed adequately. Concludingly I'd like to recommend publication of this manuscript in IJERPH.
Author Response
Thank you very much for the positive comments.
Reviewer 2 Report
Dear Authors,
I rechecked the manuscript. The introduction is better read than the last one, and the purpose is a little easier to understand.
However, we still have the following concerns.
・No ethics number is indicated.
・I believe that there should be a section on statistics. I think it would make the article easier to read if you summarize and describe what statistical methods are used to confirm what.
・non-obese (body mass index < 40) 【P7,L145】
According to the WHO definition, a BMI of 30 or more is obesity, but why did the study include subjects with less than 40? In spite of this, the mean was 23.5.
In addition, please tell us the generalization of the cluster analysis by BMI in this study.
・The figure is not well written. For example, in Figure 22-35, it is not clearly stated what a and b indicate, which is unfriendly.
・Consideration should be taken into account as the discussion is an explanation of the results.
・I think the conclusion should be a simple statement of the main points of the results of this study.
Author Response
Thank you for the valuable comments.
Responses are in the attached file.
